# Clinical Presentation of Left Ventricular Noncompaction Cardiomyopathy and Bradycardia in Three Families Carrying *HCN4* Pathogenic Variants

**DOI:** 10.3390/genes13030477

**Published:** 2022-03-08

**Authors:** Agata Paszkowska, Dorota Piekutowska-Abramczuk, Elżbieta Ciara, Alicja Mirecka-Rola, Monika Brzezinska, Dorota Wicher, Grażyna Kostrzewa, Jędrzej Sarnecki, Lidia Ziółkowska

**Affiliations:** 1Department of Cardiology, The Children’s Memorial Health Institute, 04-730 Warsaw, Poland; a.paszkowska@ipczd.pl (A.P.); a.mirecka-rola@ipczd.pl (A.M.-R.); m.brzezinska@ipczd.pl (M.B.); 2Department of Medical Genetics, The Children’s Memorial Health Institute, 04-730 Warsaw, Poland; d.abramczuk@ipczd.pl (D.P.-A.); e.ciara@ipczd.pl (E.C.); d.wicher@ipczd.pl (D.W.); 3Department of Medical Genetics, Medical University of Warsaw, 02-106 Warsaw, Poland; grazyna.kostrzewa@wum.edu.pl; 4Department of Diagnostic Imaging, The Children’s Memorial Health Institute, 04-730 Warsaw, Poland; j.sarnecki@ipczd.pl

**Keywords:** left ventricular noncompaction, cardiomyopathy, children, sinus bradycardia, *HCN4* molecular variant, late gadolinium enhancement

## Abstract

Background: Left ventricular noncompaction (LVNC) is a genetically and phenotypically heterogeneous cardiomyopathy in which myocardium consists of two, distinct compacted and noncompacted layers, and prominent ventricular trabeculations and deep intertrabecular recesses are present. LVNC is associated with an increased risk of heart failure, atrial and ventricular arrhythmias and thromboembolic events. Familial forms of primary sinus bradycardia have been attributed to alterations in *HCN4*. There are very few reports about the association between *HCN4* and LVNC. The aim of our study was to characterize the clinical phenotype of families with LVNC and sinus bradycardia caused by pathogenic variants of the *HCN4* gene. Methods: From March 2008 to July 2021, we enrolled six patients from four families with diagnosed isolated LVNC based on the clinical presentation, family history and echocardiographic and cardiovascular magnetic resonance (CMR) evidence of LVNC. Next generation sequencing (NGS) analysis was undertaken for the evaluation of the molecular basis of the disease in each family. Results: A total of six children (median age 11 years) were recruited and followed prospectively for the median of 12 years. All six patients were diagnosed with LVNC by echocardiography, and five participants additionally by CMR. The presence of late gadolinium enhancement (LGE) was found in three children. Sinus bradycardia and dilation of the ascending aorta occurred in five studied patients. In four patients from three families, the molecular studies demonstrated the presence of rare heterozygous *HCN4* variants. Conclusion: (1) The *HCN4* molecular variants influence the presence of a complex LVNC phenotype, sinus bradycardia and dilation of the ascending aorta. (2) The *HCN4* alteration may be associated with the early presentation of clinical symptoms and the severe course of the disease. (3) It is particularly important to assess myocardial fibrosis not only within the ventricles, but also in the atria in patients with LVNC and sinus bradycardia.

## 1. Introduction

Left ventricular noncompaction (LVNC) is a genetically and phenotypically heterogeneous cardiomyopathy in which myocardium consists of two distinct compacted and noncompacted layers, and prominent ventricular trabeculations and deep intertrabecular recesses are present. The frequency of LVNC has increased and it has been reported as comprising 9% of all childhood primary cardiomyopathies in recent studies [1,2]. This form of cardiomyopathy may be sporadic or familial. LVNC is associated with an increased risk of heart failure, atrial and ventricular arrhythmias and systemic thromboembolic events, both in pediatric and adult populations [3,4]. The spectrum of arrhythmias is very wide, with the most common one being ventricular or supraventricular arrhythmias and the Wolff–Parkinson–White (WPW) syndrome. Less frequently, bradyarrhythmias, including sinus bradycardia, sick sinus syndrome and varying degrees of atrioventricular (AV) block, including complete heart block, were observed in individuals with LVNC [5,6,7,8].

In the literature, there have been reports of children with LVNC in association with sinus bradycardia, suggesting the possibility that the two diseases may have a common cause [9,10]. Although familial forms of primary sinus bradycardia are recognized, they have been attributed to molecular variants in *HCN4, SCN5A* and *ANK2*. However, there are very few reports in the literature on the association between *HCN4* dysfunction and LVNC [11].

The aim of our study was to characterize the clinical phenotype of families with LVNC and sinus bradycardia caused by the *HCN4* gene alterations.

## 2. Materials and Methods

### 2.1. Study Patients

From March 2008 to July 2021, we prospectively enrolled 6 patients from 4 families with diagnosed LVNC hospitalized in the Department of Cardiology of the Children’s Memorial Health Institute. The inclusion criteria were age <18 years at the time of LVNC diagnosis based on the clinical presentation (history, symptoms, sinus bradycardia in electrocardiogram and familial occurrence of LVNC) and echocardiographic evidence of isolated LVNC. The echocardiographic criteria for LVNC diagnosis were: (1) presence of a two-layer structure with a compacted (C) and noncompacted (NC) myocardial layer of trabecular meshwork with deep endomyocardial spaces; (2) maximal end-systolic ratio of NC/C layers of >2; (3) color Doppler evidence of deep perfused intertrabecular recesses [12]. Only patients with sinus bradycardia diagnosed according to the criteria recommended in the literature were included in the study [13,14]. The exclusion criteria from the study were the presence of congenital heart disease, other forms of cardiomyopathy or neuromuscular disorders.

The Institutional Ethics Committee approved this study. Informed consent was obtained from all individual participants included in the study.

### 2.2. Data Collection

Patients’ demographics, clinical symptoms, family history of cardiomyopathies and sudden cardiac death (SCD), as well as the results of echocardiography, 12-lead resting ECG with assessment of heart rate (HR), 24 h Holter electrocardiography (ECG) with analysis of minimal, maximal, average HR, occurrence of sinus pauses (RR pauses) >2 s and cardiovascular magnetic resonance (CMR) were collected. Information on clinical symptoms, such as chest pain, arrhythmias, episodes of syncope and past thromboembolic events, were recorded, and NYHA functional class was evaluated. Serum NT-proBNP level was analyzed in all patients. Each patient underwent genetic blood tests for evaluation of the molecular basis of the disease.

### 2.3. Echocardiographic Studies

Echocardiography, including two-dimensional, Doppler and M-mode imaging, was performed at rest, as previously described [15], using Philips Epiq7 (Philips Medical Systems, Bothell, WA, USA). Echocardiographic images included parasternal long- and short-axis and apical two-, three- and four-chamber views. LVNC was diagnosed based on Jenni’s criteria, when LV NC/C ratio was ≥2.0, when measured in the parasternal short-axis view in end-systolic phase, below the papillary muscle [12]. Echocardiographic measurements included LV end-diastolic diameter (LVEDd) and LV ejection fraction (LVEF), assessed using the Simpson’s method. The results were referenced against the available pediatric normative values [16,17]. LV enlargement was diagnosed when z-score was >2; LV systolic function impairment was defined as LVEF < 55%. Measurements of the ascending aorta were taken in the parasternal long-axis view and indexed to the patient’s BSA. Ascending aorta dilation was defined as z-score > 2 [18].

### 2.4. Cardiovascular Magnetic Resonance Imaging

In all patients without contraindications, CMR studies were performed, utilizing a 1.5 T magnetic resonance scanner (Magnetom AvantoFit, Siemens, Erlangen, Germany), as previously described [19]. Steady-state free precession (SSFP) cine images of the heart were acquired at breath-hold, in the standard 2-, 3- and 4- chamber views and in short-axis, with ≥25 phases/cardiac cycle. The studies were analyzed on a dedicated diagnostic workstation using CVi42 (Circle, Calgary, AB, Canada) software. LVNC was diagnosed based on Petersen’s criteria [20] if NC/C ratio at end-diastole was >2.3. LV volumes and EF were assessed semi-automatically based on the short-axis images with papillary muscles excluded from the volumes and included in the myocardial mass [21]. The results were referenced against the multicenter pediatric CMR normative values [22]. Late gadolinium enhancement (LGE) images were obtained 10–15 min after intravenous administration of 0.1 mmol/kg of gadobutrol (Gadovist, Bayer, Berlin, Germany) and were visually evaluated for areas of myocardial hyperintensity visible in two distinct planes.

### 2.5. Molecular Studies

DNA samples were obtained from the peripheral blood by automatic isolation with MagCore Nucleic Acid Extractor HF16Plus (RBC Bioscience, New Taipei City, Taiwan). Candidate gene sequencing was performed on the HiSeq 1500 platform (Illumina, San Diego, CA, USA) using the TruSight One Sequencing Panel (Illumina) in Family 1 and the original CMHI pediatric panel of 1000 clinically relevant genes (Roche) in the remaining families. A set of 222 and 165 cardiomyopathy-associated genes were analyzed, respectively (Appendix A). Study protocol has been described in detail in our previous manuscript [23]. 

Analysis and prioritization of discovered variants considered: (1) the minor allele frequency (MAF) determined with the Genome Aggregation Database (gnomAD, https://gnomad.broadinstitute.org/ (accessed on 14 January 2022)), Exome Variant Server (EVS, https://evs.gs.washington.edu/EVS/ (accessed on 14 January 2022), UK10K Project (https://www.uk10k.org/ (accessed on 14 January 2022)) and in-house database comprising >5000 individuals (Department of Medical Genetics; DMG); (2) the pathogenicity of the variants, using in silico prediction algorithms: CADD, SIFT, MutationTaster, PolyPhen2_HDIV, PolyPhen2_HVAR, MutationAssessor, MetaSVM, MetaLR and FATHMM; (3) phenotypic descriptions in Online Mendelian Inheritance in Man (OMIM, https://www.ncbi.nlm.nih.gov/omim (accessed on 7 January 2022)), the Human Gene Mutation Database (HGMD, http://www.hgmd.cf.ac.uk (accessed on 7 January 2022)), ClinVar (https://www.ncbi.nlm.nih.gov/clinvar/ (accessed on 14 January 2022)) and Pubmed. 

All variants were validated by Sanger sequencing and reported with the *HCN4* NM_005477.3 (NP_005468.1) reference sequences. Sanger sequencing was also used for parental segregation analysis.

## 3. Results

**Baseline characteristics.** A total of six children from four families were enrolled and followed prospectively for the median of 12 (6–13) years. The median age of patients was 11 (6–16) years. Clinical characteristics of the study patients are presented in Table 1.

In echocardiography, the median NC/C ratio in all patients was 2.29 (2.13–2.40), which met Jenni’s criteria for the diagnosis of LVNC. Dilation of the ascending aorta exceeding the standard indexed to BSA was found in five study participants. In five of the six children, CMR imaging was performed to further assess cardiac morphology and function and to screen for features of myocardial fibrosis. In one child, CMR was not performed due to his severe clinical condition, hemodynamic instability and low body weight. Among the five patients who underwent CMR, the diagnosis of LVNC was confirmed in all according to Petersen’s criteria, with a median of NC/C ratio of 3.73 (3.14–4.25). All patients studied were diagnosed with sinus bradycardia in accordance with the recommendations in the literature [13,14].

**Characteristics of families.** The pedigrees of individual families are presented in Figure 1.

### 3.1. Family 1

Two sisters, one 2 years old and the other 6 years old, diagnosed with LVNC, were admitted to our institute for cardiologic evaluation. Family history revealed that their grandfather (father’s father) was diagnosed with sinus bradycardia and a pacemaker was implanted. However, his echocardiographic examination is not known, and we cannot state whether there were LVNC features. The younger girl (F I-1) diagnosed with LVNC at 3 months of age and progressive heart failure was admitted to the intensive care unit with symptoms of severe heart failure (NYHA group IV) and multi-organ injury. Upon echocardiography, the NC/C ratio was 3.34 and LV size was normal, but the LV systolic function was reduced. In addition, there was a dilation of the ascending aorta. In the resting ECG recording, the heart rate was 130 beats per minute (bpm), whereas the average daily rhythm in the 24 h Holter study was 90 bpm (the child was treated with an infusion of catecholamines). In the course of the disease, the child developed ischemic cerebral stroke. The girl died due to the mechanism of sinus bradycardia. The second girl (F I-2) underwent cardiological examinations after the diagnosis of LVNC in her sister. Clinical symptoms included dyspnea and fatigue on effort (NYHA class II). The resting ECG showed sinus bradycardia (67 bpm), and, in the 24 h Holter monitoring, the average daily rhythm was 64 bpm. Echocardiography revealed LVNC features with a NC/C ratio of 2.35 and dilation of the ascending aorta. In CMR, the NC/C ratio was 4.60 and no LGE was found. In both imaging studies, the LV size and function were within the normal range. ACE inhibitors and spironolactone were included in the treatment.

NGS analysis performed in this family in the younger girl revealed the heterozygous known *HCN4* molecular variant c.1444G>A p.Gly482Arg that was subsequently confirmed in her affected sister and her father. Nine of the used in silico algorithms (CADD, MetaSVM, MutationAssessor, Polyphen2_HDIV, Polyphen2_HVAR, MetaLR, SIFT, FATHMM, MutationTaster) predicted this change as deleterious. The c.1444G>A variant was not found in the utilized frequency population databases (gnomAD, EVS, UK10K) or in our DMG cohort. It was reported as a disease-causing variant in sinus bradycardia and myocardial noncompaction cases in HGMD (ID: CM159293), and also as pathogenic/likely pathogenic in Brugada syndrome patients in ClinVar (ID: 197253). 

### 3.2. Family 2

An 11-year-old girl (F II-3) was admitted for cardiac diagnostics due to observed sinus bradycardia and a positive family history of LVNC. Her mother had been diagnosed with LVNC and syncope, and a cardioverter-defibrillator (ICD) was implanted. In addition, her brother’s daughters were also diagnosed with LVNC. During hospitalization, the patient did not report any complaints and did not present any symptoms of heart failure (NYHA class I) and bradycardia. The ECG showed sinus bradycardia of 60 bpm, whereas the average 24 h heart rhythm in the Holter examination was 60 bpm. Echocardiography showed a NC/C ratio of 2.22, and the size of the ascending aorta was within the normal range. CMR examination confirmed the diagnosis of LVNC, and the NC/C ratio was 3.00, but no LGE was found. In both of the above studies, the LV size and systolic function were normal. Pharmacological treatment was not started.

The heterozygous known *HCN4* molecular variant c.1454C>T p.Ala485Val was identified in the proband and further confirmed in her mother. Eight in silico algorithms (CADD, MetaSVM, Polyphen2_HDIV, Polyphen2_HVAR, MetaLR, SIFT, FATHMM, MutationTaster) predicted this change as deleterious. The c.1454C>T variant was not found in the EVS, UK10K and DMG databases, but it was noted in gnomAD with MAF < 0.00001. This variant has no ClinVar submission but appeared in HGMD as a disease-causing variant in sinus bradycardia patients (ID: CM104901). 

### 3.3. Family 3

A 17-year-old patient (F III-4) was admitted to our institute due to sinus bradycardia and a family history of bradycardia and cardiomyopathy in other family members. Family history revealed that his sister was diagnosed with LVNC and sinus bradycardia, while the other sister was also treated for bradycardia. The boy’s father was diagnosed with hypertrophic cardiomyopathy and a syncope, and he had a pacemaker implanted. The patient’s clinical symptoms included only dyspnea and fatigue on effort (NYHA class II). The electrocardiogram showed a sinus rhythm of 44 bpm (Figure 2).

The mean 24 h heart rate during the Holter test was 47 bpm. Echocardiography revealed LVNC phenotypic features with a NC/C ratio of 2.40 and dilation of the ascending aorta. In the CMR study, the NC/C ratio was 3.73 and the presence of LGE in the LV myocardial wall was found. Both echocardiography and CMR showed a normal LV size and LV systolic function. The patient was receiving treatment for heart failure (ACE inhibitor, spironolactone) and salbutamol for sinus bradycardia.

The heterozygous known *HCN4* molecular variant c.1438G>C p.Gly480Arg was found in the proband, and was also identified in his father (DNA from his sisters was not available for testing). Nine in silico algorithms (CADD, MetaSVM, MutationAssessor, Polyphen2_HDIV, Polyphen2_HVAR, MetaLR, SIFT, FATHMM and MutationTaster) predicted this change as deleterious. The c.1438G>C variant was not present in the frequency population databases. It was reported as disease-associated polymorphism in sinus bradycardia asymptomatic association with HGMD (ID: CM073122), and was classified as pathogenic in two patients with sick sinus syndrome in ClinVar (ID: 5176).

### 3.4. Family 4

The siblings, a 16-year-old girl (F IV-5) and her 11-year-old brother (F IV-6), were admitted to our Department of Cardiology due to sinus bradycardia. Family history revealed that their father had been diagnosed with sinus bradycardia and dilated cardiomyopathy (DCM); he had a pacemaker implanted. Moreover, their father’s brother also had a pacemaker implanted due to sinus bradycardia. The girl was asymptomatic, and no syncope was present. She only reported increased fatigue during exercise (NYHA class II). A nodal rhythm of approximately 39 bpm was recorded in the resting ECG. In the 24 h Holter ECG, the average daily rhythm was 55 bpm. Sick sinus syndrome (SSS) and supraventricular premature beats were diagnosed. Echocardiography revealed LVNC with a NC/C ratio of 2.06 and dilation of the ascending aorta. In the CMR examination, the NC/C ratio was 3.91, and foci of LGE were found both in the myocardial wall of the left ventricle and in the wall of the left atrium (Figure 3).

In both echocardiography and CMR, the LV was enlarged, with a borderline systolic function. The girl received treatment for heart failure (ACE inhibitor, spironolactone) and salbutamol due to sinus bradycardia.

The second sibling, the boy, had no symptoms of sinus bradycardia or syncope. Clinical symptoms included dyspnea and fatigue on effort (NYHA class II). The heart rate on the ECG was 56 bpm, with a mean daily heart rhythm on the 24 h Holter study of 50 bpm. The patient was diagnosed with sick sinus syndrome and premature supraventricular beats, without episodes of supraventricular tachycardia. Echocardiography revealed typical LVNC features, with a NC/C ratio of 2.13 and LV enlargement with normal systolic function. There was also a dilation of the ascending aorta. CMR confirmed the diagnosis of LVNC (the NC/C ratio was 3.27), whereas the LV size and systolic function were normal. Like his sister, the boy was found to have LGE in the LV myocardial wall and in the left atrium. The patient is treated with medications such as ACE inhibitors and salbutamol. 

A molecular analysis performed on the girl revealed no candidate pathogenic/likely pathogenic variant in the studied cardiomyopathy panel.

## 4. Discussion

Left ventricular noncompaction cardiomyopathy is a genetically and phenotypically heterogeneous myocardial disease characterized by multiple prominent trabeculations and deep intertrabecular recesses. It is a relatively new, yet rare, clinical entity, but nevertheless the third most common cardiomyopathy in childhood, and is associated with congestive heart failure, arrhythmias, atrioventricular conduction disorders, sudden cardiac death and/or thromboembolic events. However, patients with isolated LVNC can also be asymptomatic and diagnosed after an abnormal ECG, echocardiogram or family screening [24,25]. 

In the literature reports, the relationship between LVNC and sinus bradycardia has been described in approximately 55% of cases, although the basis of conduction disturbances remains unclear [8,26]. The results of other studies demonstrated that the cause of sinus bradycardia in patients with LVNC may be the presence of a pathogenic variant in the *HCN4* gene [11,27,28].

Molecular variants in *HCN4* are thought to predominantly underlie sinoatrial node (SAN) disorders because *HCN4* expression is mostly limited to the cardiac conduction system, and especially the SAN area [27]. HCN4 is the most prominent channel in the sinoatrial node, being a major determinant of the cardiac pacemaker current (If) and playing a crucial role in the automaticity of the sinus node through the generation of a slow diastolic depolarization during phase four of the cardiac action potential. Thus, this is a crucial channel for appropriate pacemaker activity and conduction system function [27,29]. Cardiac pacemaker current channel malfunction due to *HCN4* dysfunction may result in cardiac function disorders. They have been reported mainly in patients with sick sinus syndrome (SSS) and Brugada syndrome. However, in recent years, a broad spectrum of conditions, including sinus bradycardia, sinus tachycardia, atrial fibrillation, atrioventricular block, idiopathic ventricular tachycardia, left ventricular noncompaction, myocardial infraction, sudden infant death syndrome, arrhythmogenic right ventricular cardiomyopathy, dilation of the aorta and chronotropic incompetence, have been noted [27,30]. 

The epidemiology of *HCN4* pathogenic variants in LVNC patients is not fully determined, with limited data concerning, in particular, the pediatric population. This can be due to the rare occurrence of childhood LVNC, lack of widespread routine genetic testing and lack of *HCN4* in the applied gene panels. Van Waning et al. [31] collected data from papers published between January 1999 and March 2018 with clinical and molecular characteristics of 561 LVNC cases (adults and children) and found twenty-two subjects (4%) who carried *HCN4* alterations causative for the disease. *HCN4* variants accounted for 10% of (likely) pathogenic variants identified in a cohort of 95 adults diagnosed with LVNC by Richard et al. [32], as well as in a similar group reported on by Cambon-Viala et al. [33]. In a recent study, Hirono et al. [34] found two children with *HCN4*-related LVNC among 206 patients aged less than 16 years. 

Milano et al., were the first to describe the coexistence of LVNC, sinus bradycardia and molecular variants in the *HCN4* gene and emphasized the role of their familial occurrence [11]. Although there are reports showing significant phenotypic heterogeneity in pathogenic *HCN4* gene variants, only patients who have the LVNC phenotype have been described. Moreover, the detection of pathogenic variants in patients with LVNC identifies cases (and their family members) who are at risk of developing adverse cardiac events, including arrhythmias [35].

To date, at least 50 (likely) pathogenic *HCN4* molecular variants have been collected in the Human Gene Mutation Database Professional 2021v.3. All molecular variants identified in this study are known changes that affect highly evolutionary conserved amino acids located in close proximity with the essential pore-forming domain of the protein (Figure 4). 

These variants have been reported previously, mainly in adults with a complex phenotype of LVNC and sinus bradycardia and myocardial noncompaction [28,32,33,36,37], whereas pediatric presentations are rare. Familial forms of SSS caused by *HCN4* alterations rarely manifest themselves clinically in childhood and are mainly recognized after adolescence. Only a few literature descriptions of teenage patients manifested with a complex phenotype of sinus bradycardia and LVNC can be found in literature. Ishikawa et al., described a case of a 13-year-old girl with significant sinus bradycardia requiring permanent pacing at the age of 17 years [37]. Millat et al. [28] reported on a family with three sisters (12, 18 and 24 years old) who presented with a similar phenotype: sinus bradycardia in combination with LVNC. In our study, we present early onset of clinical symptoms of the disease. Both siblings from Family 1 were diagnosed before 6 years of age, with severe multi-organ dysfunction and ischemic cerebral stroke resulting in death at 2 years in one of them.

Different base substitution at the same *HCN4* nucleotide position (c.1438, c.1444 and c.1454) leading to the same or various amino acid changes has been reported in multiple studies [11,29,32,33], highlighting the important role of this region for the proper functioning of the *HCN4* channel. The functional effect of these variants has been studied extensively, demonstrating that the p.Gly480Arg, p.Gly482Arg and p.Ala485Val variants severely abolish pacemaker currents, with a hyperpolarization shift in the voltage dependence of activation [11,29,38,39]. Recent literature reports indicate that the molecular etiology of LVNC is detected in approximately 40% of cases [40]. As in the reports of other authors [11,28], in our patients, a significant relationship between LVNC, sinus bradycardia and *HCN4* molecular variants has been demonstrated. The molecular etiology of heart abnormalities was confirmed in three out of four families (a total of four patients). It should be emphasized that, in Family 4, despite the typical course of LVNC and sinus bradycardia, no molecular cause was identified. 

A very important aspect of the clinical evaluation of our patients is to demonstrate the presence of dilation of the ascending aorta. There are only a few studies showing an association of the *HCN4* gene alterations with dilation of the ascending aorta. As reported by Vermeer et al., evidence was provided that dilation of the ascending aorta also constitutes a part of the clinical spectrum of *HCN4* changes. Moreover, these authors demonstrated that *HCN4* variant carriers have aortic dilation with age, which undoubtedly influences the prognosis and cardiac care in this group of patients [41].

According to the literature, the presence of LGE in the LV myocardium is found in 40% of LVNC patients [42]. In our study, out of five patients who underwent CMR examination, three (60%) children showed LGE in the LV myocardium and in the left atrium. The presence of LV fibrosis is a strong independent predictor of poor prognosis, not only in LVNC but also in another form of childhood cardiomyopathy, i.e., hypertrophic cardiomyopathy [43]. The results of published studies show that the prognosis worsens in patients with LVNC and the presence of LGE and/or global impairment of LV systolic function [44,45]. These authors also emphasize that only hypertrabeculation meeting current LVNC diagnostic criteria has no significant impact on patient prognosis, but, when the LVNC pattern is associated with a DCM-like phenotype and/or LGE, the prognosis deteriorates significantly. Among our three patients with LVNC and the presence of LGE, two children had LV enlargement requiring further regular cardiological control. 

The literature presents the results of a study showing that atrial fibrosis, quantified with CMR-LGE, is an important predictor of clinically significant sinus node dysfunction and sinus bradycardia requiring pacemaker implantation [46]. It should be especially emphasized that, in our group of three children with LVNC and LGE, two had LGE both in the LV wall and within the left atrial wall. These patients were diagnosed with sinus node dysfunction and sinus bradycardia, and further investigation was scheduled prior to qualification for permanent pacing implantation. 

In summary, a systematic search for *HCN4* molecular variants should be undertaken in patients with such a complex clinical picture. Additionally, we emphasized the role of CMR in LVNC, not only regarding diagnosis but also risk stratification and future medical treatment. This is important for everyday clinical practice and for further patient care, as it enables targeted treatment and further management.

## 5. Conclusions

The *HCN4* molecular variant influences the presence of a complex LVNC phenotype, sinus bradycardia and dilation of the ascending aorta;*HCN4* alteration may be associated with the early presentation of clinical symptoms and severe course of the disease;It is particularly important to assess myocardial fibrosis not only within the ventricles, but also in the atria in patients with LVNC and sinus bradycardia.

## Figures and Tables

**Figure 1 genes-13-00477-f001:**
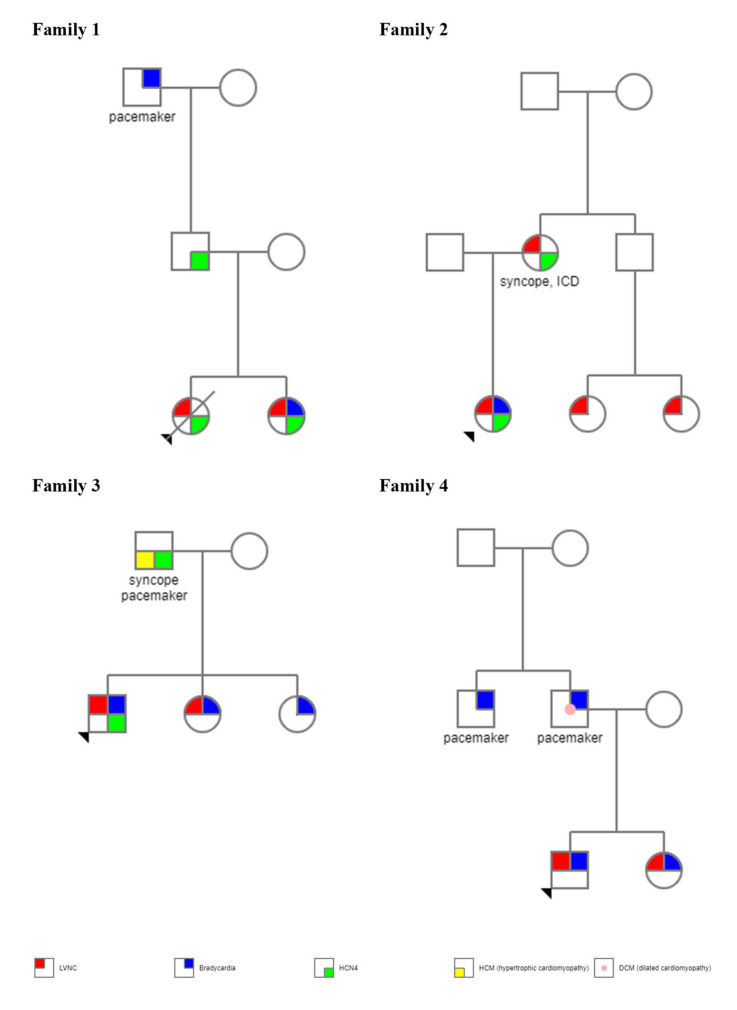
Pedigree of four families with LVNC and sinus bradycardia. Circle—females; square—males; arrow—proband. In red—patients affected by left ventricular non-compaction, in blue—bradycardia, in green—*HCN4* molecular variant, in yellow—hypertrophic cardiomyopathy, in pink circle—dilated cardiomyopathy. A strikethrough field—death.

**Figure 2 genes-13-00477-f002:**
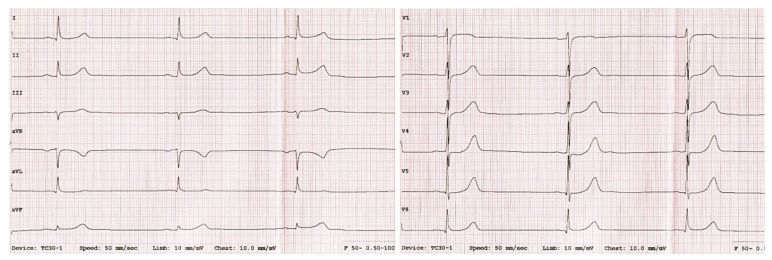
ECG recording shows sinus bradycardia of 44 bpm in a 17-year-old patient.

**Figure 3 genes-13-00477-f003:**
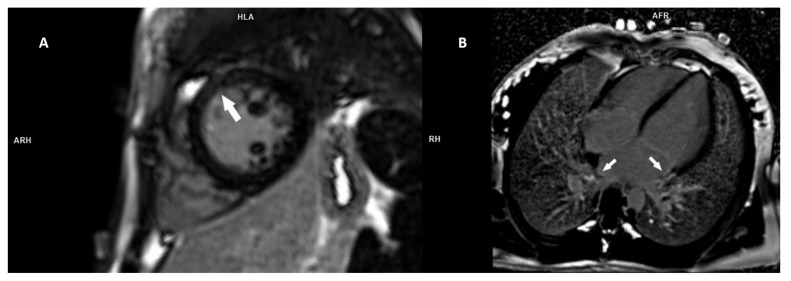
Late gadolinium enhancement (LGE) PSIR images showing midwall LGE in the mid anterior and anteroseptal segments of the left ventricle (**A**)—short-axis view and LGE in the left atrium (**B**)—4-chamber view, in a 16-year-old female patient.

**Figure 4 genes-13-00477-f004:**
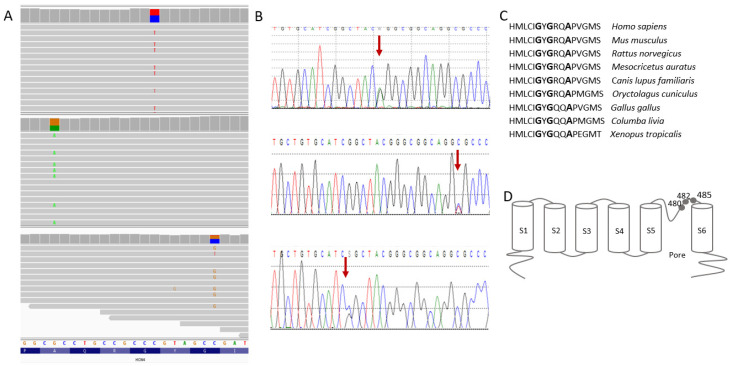
Heterozygous molecular variants c.1444G>A, c.1454C>T, and c.1438G>C identified in this study: (**A**) NGS results displayed in Integrative Genomics Viewer v.2.10.0 (http://software.broadinstitute.org/software/igv/ (accessed on 14 January 2022)); (**B**) Sanger sequencing electropherograms. Arrows indicate an altered nucleotide; (**C)** Multiple alignment of HCN4 region of interest, showing amino acid residues conserved between species (bold font); (**D**) schematic topology of HCN4 with the altered amino acids indicated. The S1–S6 blocks denote structural segments, with the pore-forming region comprising amino acids in positions 465–486.

**Table 1 genes-13-00477-t001:** Baseline characteristics of the study patients.

Family-Patient	Age (yrs)	NYHA Class	Syncope	Embolicevents	NTpro-BNP pg/mL	HR (ECG) bpm	Min HR (Holter) bpm	Max HR (Holter) bpm	Average HR (Holter) bpm	RR Pauses > 2 s	NC/C (Echo)	LVEF (Simpson) %	LVDd mm (z-Score)	Ao asc mm (z-Score)	LVEF (CMR) %	LV EDV/BSA (z-Score)	NC/C (CMR)	LGE	Treatment
F I-1	2	IV	0	1	N/A	130	61	133	90	0	3.34	59	28.4 (+0.5)	21 (+4.2)	N/A	N/A	N/A	N/A	1
F I-2	6	II	0	0	110.80	67	36	129	64	0	2.35	64	35.6 (−0.5)	25.4 (+4.0)	71.0	73 (0.7)	4.60	0	1
F II-3	11	I	0	0	<5.00	60	37	147	60	0	2.22	70	46.3 (+0.9)	19.4 (−0.7)	71.3	69.6 (1.2)	3.00	0	0
F III-4	17	II	0	0	32.95	44	32	119	47	0	2.40	71	41.1 (+0.8)	35.2 (+5.9)	63.7	125.8 (−1)	3.73	1	1
F IV-5	16	II	0	0	91.10	39	40	92	55	0	2.06	57	59.5 (+2.5)	36.9 (+4.2)	54.2	122.1 (3.7)	3.91	1	1
F IV-6	11	II	0	0	19.04	56	43	79	50	0	2.13	62	47.3 (+2.3)	34 (+4.7)	70.2	114.1 (1.2)	3.27	1	1

F I-IV—families, 1–6—family members, NYHA—New York Heart Association, yrs—years, NTproBNP—N-terminal pro-brain natriuretic peptide, HR—heart rate, bpm—beat per minute, ECG—electrocardiogram, Min—minimal, Max—maximal, RR pauses—sinus pauses, s—seconds, NC/C—noncompacted to compacted myocardial layer ratio, LVEF—left ventricular ejection fraction, LVDd—left ventricular diastolic diameter, Ao asc—ascending aorta, CMR— cardiovascular magnetic resonance, BSA—body surface area, LV EDV—left ventricular end-diastolic volume, LGE—late gadolinium enhancement, N/A—data are not available, ‘’0′’—were not present, “1”—were present.

## Data Availability

The data presented in this study are available on request from the corresponding author.

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
