# Peer review of "Clinical Presentation of Left Ventricular Noncompaction Cardiomyopathy and Bradycardia in Three Families Carrying HCN4 Pathogenic Variants"

_genes, 2022, doi:10.3390/genes13030477_

Round 1
Reviewer 1 Report
To the authors.
I have read with extreme interest the article by Agata Paszkowska et al. entitled “Clinical presentation of left ventricular noncompaction cardiomyopathy and bradycardia in three families carrying HCN4 pathogenic variants”.
When the World Health Organization (WHO) published the first definition of cardiomyopathy in the 80s, noncompaction cardiomyopathy (NCCM) was not yet known.
There is a broad spectrum of clinical presentations with primarily heart failure symptoms, different forms of arrhythmias and thromboembolic events. Between the onset of symptoms and the diagnosis there can be a delay of up to 3 to 4 years. Initially, the term isolated NCCM was used for cases without congenital or other
structural heart defects. Today, some groups use the term for cases with areas of LVNC
and normal LV function.
NCCM in the paediatric and adult population probably has a different background. The National Australian Childhood Cardiomyopathy study documented 9.2% of the primary cardiomyopathies in children younger than 10 years of age as NCCM. Arrhythmias are frequent in NCCM, ventricular as well as supraventricular arrhythmias. Thromboembolic events mainly occur in patients with NCCM and atrial fibrillation.
Stasis of blood flow can appear in the deep intertrabecular recesses notably in reduced LV function. Neurologic departments occasionally diagnose NCCM in patients with otherwise not explained stroke. In cohorts with NCCM, a percentage of about 10–15% suffer from stroke Several congenital heart defects were described with NCCM. A patient with pulmonic valve atresia was described in 1964 and patients with Ebstein anomaly in 2005. Friedberg described a patient with atrial isomerism. Shunt defects such
as ventricular septal defect, atrial septal defect and patent ductus arteriosus Botalli were described as well. Several authors described noncompaction in patients with congenital heart disease, in Epstein anomaly, subaortic VSD, bicuspid aortic valve and tetralogy of Fallot. This in fact strengthens the need for a comprehensive echocardiographic evaluation of any patient with newly diagnosed NCCM to rule out congenital heart disease. In about 50% of children with concomitant face dysmorphisms or a neutropenia (Barth syndrome), cardiomyopathy with and without noncompaction was described.
Under this light, the present paper strengthen the clinical characteristics of NCCM but also adds: a) the rarer presentation of the aortic dilation in addition to the common features and the b) association –as causative gene- to a classic ion-channel gene thus rendering the paper more intriguing.
For my point of view it is well written and presented.
I would only specify the term LGE in the abstract.
Author Response
Thank you for your review and the feedback on our research.
According to your suggestion the abbreviation LGE was defined in the abstract.

Reviewer 2 Report
Authors present cases of six children from three families, with a rare cardiac disorder, left ventricular non-compaction cardiomyopathy (LVNC), and examine its relationship with HCN4 mutation, which is also associated with bradycardia and dilatation of the ascending aorta.
Introduction section discusses in short the pathology and prevalence of LVNC as well as possible genetic background of the disease, HCN4 being considered as one of the potential causes.
Material and methods section presents the study group, imaging modalities used for the initial diagnosis and molecular methods used to identify the mutation. Diagnostic modalities are appropriate to the disorder.
Results section contains baseline characteristics of the group and descriptions of individual cases and family relations in sufficient detail.
Discussion presents available literature data, which is quite limited due to the rare nature of the condition and lack of widespread genetic testing. Authors discuss the process of diagnosis and prognostication in LVNC, concluding that the presence of HNC4 mutation is likely to be related to a complex phenotype and poor outcome of LVNC.
Even though the case report contains only 6 cases (not surprising, due to the rarity of the condition), long follow-up period (median of 12 years) and extensive diagnostics certainly add greatly to the strength of this study.
It is a valuable and important publication, which will certainly add to the body of knowledge in this rare condition.
Minor spell and grammar check is required.
Author Response
Thank you for your review. The manuscript was reread and the found spelling and grammatical errors were corrected.

Reviewer 3 Report
The authors followed 6 LVNC patients from 4 families, The authors sequenced the targeted genes and found Hcn4 was mutated in these patients. They characterize the patients with ECHO and CRM. The study was well designed and the conclusion was supported by the data. Only minor changes are requested.
Labeling in figure 1 is not clear.
LGE in the abstract was not explained.
Discuss if Hcn4 mutation in the animal models display LVNC and aortic dilation.
Author Response
Thank you for your review and the valuable comments on our paper.
According to your suggestion the abbreviation was defined in the abstract. The legend in Figure 1 was in low resolution, it was exchanged for a higher resolution image.
In the literature there are few studies reported animal models with mutated Hcn4 gene including mutants with recurrent sinus pauses (Stieber J et al., 2003, Herrmann et al., 2007; Harzheim et al., 2008; Hoesl et al., 2008), and mutants that exhibit a severe phenotype with pronounced bradycardia and atrioventricular block with heart arrest and embryonic or early death (Alig et al., 2009; Baruscotti et al., 2011).
The papers mostly demonstrated an altered cAMP level, influencing a basal HCN channel function, a strongly diminished If in myocytes and confirmed deleterious effect of the particular molecular variant and the primary role of HCN4 in cardiac function.
We have not identified reports on LVNC or aortic dilation in animal models with mutated Hcn4 gene.
